# A Closer Look at Time Steps is Worthy of Triple Speed-Up for Diffusion Model Training

## Abstract

Training diffusion models is always a computation-intensive task. In this paper, we introduce a novel speed-up method for diffusion model training, called *SpeeD*, which is based on a closer look at time steps. Our key findings are: i) Time steps can be empirically divided into acceleration, deceleration, and convergence areas based on the process increment. ii) These time steps are imbalanced, with many concentrated in the convergence area. iii) The concentrated steps provide limited benefits for diffusion training. To address this, we design an asymmetric sampling strategy that reduces the frequency of steps from the convergence area while increasing the sampling probability for steps from other areas. Additionally, we propose a weighting strategy to emphasize the importance of time steps with rapid-change process increments. As a plug-and-play and architecture-agnostic approach, *SpeeD* consistently achieves $3\times$ acceleration across various diffusion architectures, datasets, and tasks. Notably, due to its simple design, our approach significantly reduces the cost of diffusion model training with minimal overhead. Our research enables more researchers to train diffusion models at a lower cost.

## 1 Introduction

Training diffusion models is not usually affordable for many researchers, especially for ones in academia. For example, DALL·E 2 (OpenAI, 2023) needs 40K A100 GPU days and Sora (OpenAI, 2024) at least necessitates 126K H100 GPU days. Therefore, accelerating the training for diffusion models has become urgent for broader generative AI and other computer vision applications.

Recently, some acceleration methods for diffusion training focus on time steps, primarily using re-weighting and re-sampling 1) Re-weighting on the time steps based on heuristic rules. P2 (Choi et al., 2022) and Min-SNR (Hang et al., 2023) use monotonous and single-peak weighting strategies according to sign-to-noise ratios (SNR) in different time steps. 2) Re-sampling the time steps. Log-Normal (Karras et al., 2022) assigns high sampling probabilities for the middle time steps of the diffusion process. CLTS (Xu et al., 2024) proposes a curriculum learning based time step schedule, gradually tuning the sampling probability from uniform to Gaussian by interpolation for acceleration as shown in Fig. 1b.

To investigate the essence of the above accelerations, we take a closer look at the time steps. The diffusion models essentially learn and estimate the *process increment* noted as $\delta_t := x_{t+1} - x_t$ at time step $t$ at training and inference phase. As illustrated in the left of Fig. 1a, we visualize the changes of mean and variance of process increment through the time steps. The time steps are categorized into three areas: acceleration, deceleration, and convergence, based on observations and theoretical analyses (Sec. 2.2) that both the changes of mean and variance initially accelerate, subsequently decelerate, and ultimately converge in a narrow interval.

One can easily find that the proportions of the three areas are imbalanced: there are a large number of time steps at convergence area, and the others are small. Another finding is that process increments at the convergence area are almost identical noise, *e.g.*, in DDPM, the distribution are nearly $\mathcal{N}(0, 2\mathbf{I})$, where $\mathbf{I}$ is the unit matrix. To further explore the characteristics of these three areas, we visualize the training loss curve in the right of Fig. 1a. The loss values from the convergence area are much lower than those from the other areas, which indicates estimating the identical noise is easy.

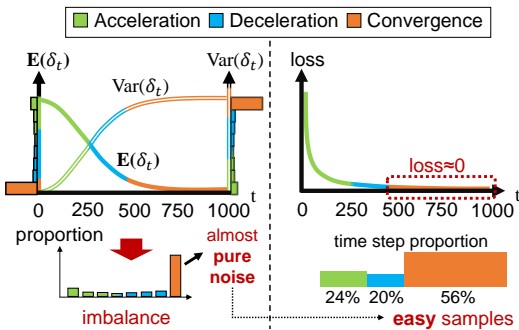
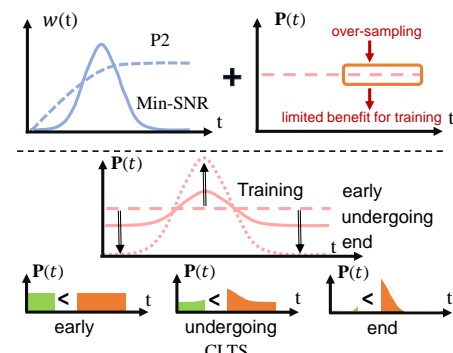

(a) Closer look at time steps: More than half of the time steps are easy-to-learn.

(b) Re-weighting and re-sampling methods can't eliminate the redundancy and under-sample issues.

Figure 1: Motivation: designing an efficient training via analyzing process increment $\delta_t$ at different time steps. (a) $\mathbf{E}(\delta_t)$ and $\mathrm{Var}(\delta_t)$ are the mean and variance of process increments $\delta_t$. Two histograms represent the proportions of the process increments at different noise levels (left) and the proportions of the time steps (right) in the three areas. The loss curve is obtained from DDPM (Ho et al., 2020) on CIFAR-10 (Krizhevsky et al., 2009). (b) $w(t)$ and $\mathbf{P}(t)$ are respectively the weighting and sampling curve. The probability of convergence area being sampled remains, while the one of acceleration is reduced faster.

Previous acceleration works have achieved promising results, but the analysis of time steps remains relatively under-explored. P2 (Choi et al., 2022) and Min-SNR (Hang et al., 2023) are two re-weighting methods, with their weighting curves across time steps as shown in Fig. 1b. They employ uniform sampling of time steps, which include too many easy samples from the convergence area during diffusion model training.

On the other hand, most re-sampling methods heuristically emphasize sampling the middle-time steps, but they do not dive into the difference between the acceleration and convergence areas. For example, CLTS (Xu et al., 2024) gradually changes the sampling distribution from uniform to Gaussian by interpolation as shown in Fig. 1b. The sampling probability of the acceleration area drops faster than the one of the convergence area. The acceleration area is still under-sampled and therefore not well-learned.

Motivated by the analyses from a closer look at time steps, we propose *SpeeD*, a novel approach that aims to improve the training efficiency for diffusion models. The core ideas are illustrated in Fig. 2. To mitigate the redundant training cost, different from uniform sampling, we design an asymmetric sampling strategy that suppresses the attendance of the time steps from the convergence area in each iteration. Meanwhile, we weight the time steps by the change rate of the process increment, emphasizing the importance of the rapid-change intervals.

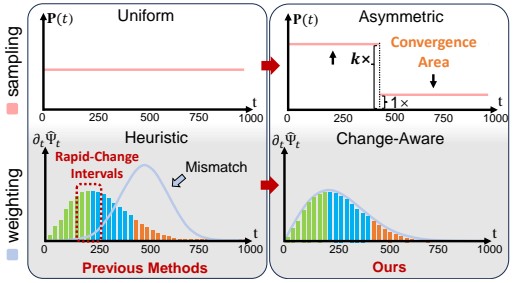

Figure 2: Core designs of SpeeD. Red and blue lines denote sampling and weighting curves.

Our approach has the following characteristics: SpeeD is compatible with various diffusion model training methods, *i.e.*, U-Net (Ronneberger et al., 2015) and DiT (Peebles & Xie, 2023) with minimal modifications. For performance, SpeeD achieves non-trivial improvements than baseline and other methods at the same training iterations. For efficiency, SpeeD consistently accelerates the diffusion training by $3\times$ across various tasks and datasets. It helps mitigate the heavy computational cost for diffusion model training, enabling more researchers to train a model at an acceptable expense. The extra time complexity of SpeeD is $\mathcal{O}(1)$, costing only seconds to reduce days of diffusion models training on datasets like FFHQ, MetFaces and ImageNet-1K. We hope this work can bring novel insights for efficient diffusion model training.

## 2 SPEEDING UP TRAINING FROM PROCESS INCREMENTS AT TIME STEPS

In this section, we first introduce the preliminaries of diffusion models and then focus on a closer look at time steps and key designs of our proposed SpeeD .

### 2.1 PRELIMINARIES OF DIFFUSION MODELS

In conventional DDPM (Ho et al., 2020; Sohl-Dickstein et al., 2015), given data $x_0 \sim p(x_0)$, the forward process is a Markov-Gaussian process that gradually adds noise to obtain a perturbed sequence $\{x_1, x_2, ..., x_T\}$,

$$q(x_t|x_{t-1}) = \mathcal{N}(x_t; \sqrt{1-\beta_t}x_{t-1}, \beta_t \mathbf{I}), q(x_{1:T}|x_0) = \prod_{t=1}^{T} q(x_t|x_{t-1}),$$

where $\mathbf{I}$ is the unit matrix, $T$ is the total number of time steps, $q$ and $\mathcal{N}$ represent forward process and Gaussian distribution parameterized by scheduled hyper-parameters $\{\beta_t\}_{t\in[T]}$. Perturbed samples are sampled by $x_t = \sqrt{\bar{\alpha}_t} \cdot x_0 + \sqrt{1-\bar{\alpha}_t} \cdot \epsilon, \epsilon \sim \mathcal{N}(0, \mathbf{I})$, where $\alpha_t = 1 - \beta_t$ and $\bar{\alpha}_t = \prod_{s=1}^{t} \alpha_s$.

For diffusion model training, the forward process is divided into pairs of samples and targeted process increments by time steps $t$, defined as $\delta_t := x_{t+1} - x_t$. The diffusion model is expected to predict the next step from the given time step. The training loss (Ho et al., 2020) for diffusion models is to predict the normalized noise. The loss highlighted with weighting and sampling modules:

$$L = \mathbb{E}_{\mu_t}[w_t||\epsilon - \epsilon_\theta(x_t, t)||^2] := \int_t w_t||\epsilon - \epsilon_\theta(x_t, t)||^2 \mathbf{d}\mu_t, \epsilon \sim \mathcal{N}(0, \mathbf{I}). \quad (1)$$

Intuitively, a neural network $\epsilon_\theta$ is trained to predict the normalized noise $\epsilon$ added at given time-step $t$. The probability of a sample being sampled in the forward process is determined by the probability measure $\mu_t$, while the weight of the loss function is determined by $w_t$ at $t^{\text{th}}$ time-step.

### 2.2 CASE STUDY ON DDPM: A CLOSER LOOK AT TIME STEPS

In DDPM, the diffusion model learns the noise added in the forward process at given $t^{\text{th}}$ time step. The noise is presented as $\epsilon$, the label in Eqn. 1, which is the normalized process increment at given time step. This label tells what the output of the diffusion model is aligning to. To take a closer look, we focus on the nature of the process increment $\delta_t$ itself to study the diffusion process $x_t \rightarrow x_{t+1}$, instead of $\epsilon$ the normalized one. According to Theorem 1 and Remark 1, based on the variation trends of process increments $\delta_t$, we can distinguish three distinct areas: acceleration, deceleration, and convergence. The detailed discussion is shown as follows:

**Theorem 1** (Process increment in DDPM). *In DDPM's setting (Ho et al., 2020), the linear schedule hyper-parameters $\{\beta_t\}_{t\in[T]}$ is an equivariant series, the extreme deviation $\Delta_\beta := \max_t \beta_t - \min_t \beta_t$, $T$ is the total number of time steps, and we have the bounds about the process increment $\delta_t \sim \mathcal{N}(\phi_t, \Psi_t)$, where $\phi_t := (\sqrt{\alpha_{t+1}} - 1)\sqrt{\bar{\alpha}_t}x_0, \Psi_t := [2 - \bar{\alpha}_t(1 + \alpha_{t+1})]\mathbf{I}$, $\mathbf{I}$ is the unit matrix:*

$$\begin{aligned}
&\textit{Upper-bound:} && ||\phi_t||^2 \leq \hat{\phi}_t||\mathbb{E}x_0||^2, && \hat{\phi}_t := \beta_{\max} \exp\{-(\beta_0 + \Delta_\beta t/2T)t\} \\
&\textit{Lower-bound:} && \Psi_t \succeq \hat{\Psi}_t \mathbf{I}, && \hat{\Psi}_t := 2 - 2\exp\{-(\beta_0 + \Delta_\beta t/2T)t\}
\end{aligned} \quad (2)$$

**Remark 1.** *The entire diffusion process can be approximated using the upper and lower bounds from Theorem 1, which we visualize as shown in Figure 3. We can observe that the diffusion process can be divided into three areas: acceleration, deceleration, and convergence. The two boundary points of these areas are denoted as $t_{\text{a-d}}$ and $t_{\text{d-c}}$ with their specific definitions and properties outlined below.*

**Definition of $t_{\text{a-d}}$.** The boundary between the acceleration and deceleration areas is determined by the inflection point in the parameter variation curves, as illustrated in Figure 3. This inflection point represents the peak where the process increment changes most rapidly. The key time-step $t_{\text{a-d}}$ between acceleration and deceleration areas satisfies $t_{\text{a-d}} = \arg\max_t \partial_t \hat{\Psi}_t$ and $\beta_{t_{a-d}} = \sqrt{\Delta_\beta/T}$ in our setting, where $\partial_t \hat{\Psi}_t = 2(\beta_0 + \Delta_\beta t/T) \exp\{-(\beta_0 + \Delta_\beta t/2T)t\}$.

**Definition of $t_{\text{d-c}}$.** The process is considered to be in the convergence area where the increments' variance is within a range. The convergence area is identified by a magnitude $r$, where $1 - 1/r$ is the ratio to the maximum variance.

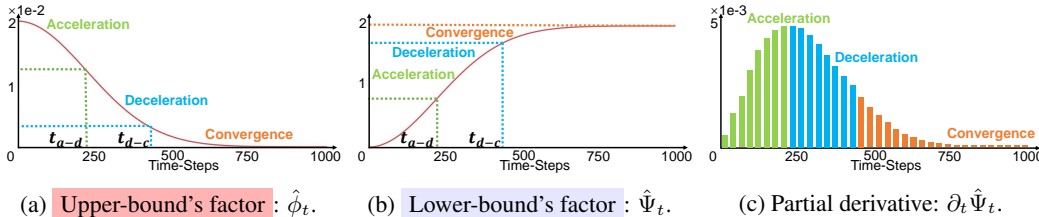

(a) Upper-bound's factor : $\hat{\phi}_t$.  (b) Lower-bound's factor : $\hat{\Psi}_t$.  (c) Partial derivative: $\partial_t\hat{\Psi}_t$.

Figure 3: Visualization of Theorem 1: three areas of acceleration, deceleration and convergence.

According to Theorem 1, the convergence area is defined as *one magnitude reduction* of the scale factor (*i.e.*, $1 \times r$), and we have the lower-bound of the magnitude $\hat{r}_t := \exp\{(\beta_0 + \Delta_\beta t/2T)t\}$ employed as the threshold selection function in Section 2.5. The time step $t$ is guaranteed to be in convergence area when $\hat{r}_t > r$, .

**Analyses.**  In the convergence area, the variations of $\delta_t$ stabilize, signifying that the process is approaching a steady state.

This area corresponds to a very large proportion of the overall time steps. On top of that, the training loss in this area is empirically low, which leads to the redundant training cost on time steps with limited benefits for training.

In the acceleration area, the variations of $\delta_t$ increase, indicating a rapid change. Conversely, in the deceleration area, the variations of $\delta_t$ decrease, reflecting a slowing down of the process. Notably, near the peak between the acceleration and deceleration areas, the process exhibits the fastest changes. These time steps only occupy a small proportion. Beyond that, the training losses in this area are empirically high. The issue is that a hard-to-learn area is even under-sampled, necessitating more sampling and training efforts.

**Takeaways.**  Based on the analyses and observations, we provide takeaways as follows:

- The samples from convergence area provide limited benefits for training. The sampling of the time step from this area should be suppressed.
- Pay more attention to the process increment's rapid-change area which is hard to learn and the corresponding time steps are fewer than the other areas.

## 2.3 GENERAL CASES BEYOND DDPM: DIVE INTO THE COMMONALITY.

This section generalize the above findings on DDPM to broader settings. The findings are about the process increments $\delta_t := x_{t+1} - x_t$, and the related differentiation of right limit $\mathbf{d}x$ in forward process. The corresponding SDE (Karras et al., 2022) and the discretization are:

$$\mathbf{d}x = x\dot{s}/s\mathbf{d}t + s\sqrt{\dot{\sigma}\sigma}\mathbf{d}w, \qquad x_t = s_t x_0 + s_t\sigma_t\epsilon, \epsilon \sim \mathcal{N}(0,\mathbf{I}),$$

where scale factor $s = s_t$ and noise standard deviation (std.) $\sigma = \sigma_t$ are the main designs related to the main factors $\hat{\phi}_t$ and $\hat{\Psi}_t$ about process increment $\delta_t$ in Theorem 1 across time steps $t$.

**Generalize Theorem 1: $s$-$\sigma$ Scheduled Process Increments.**  The generalized process increment is $\delta \sim \mathcal{N}(\Delta x_0, \Sigma\mathbf{I})$, where $\Delta := s_+ - s$ and $\Sigma := s_+^2\sigma_+^2 + s^2\sigma^2$ across $t$. $\Delta$, $\Sigma$ are continuous on $t$ without discretization, where $s_+$ and $\sigma_+$ are the right outer limits, *i.e.*, $s(t + \mathbf{d}t)$. In discretization, $\Delta_t$ and $\Sigma_t$, marked by $t$, are related to sample granularity of time step $t$. Like Theorem 1, we study the variation of process increments by $\dot{\Delta} = \dot{s}_+ - \dot{s}$ and $\dot{\Sigma} = \mathbf{m}^\top\dot{\mathbf{n}}$ where $\mathbf{m} = [s_+^2, \sigma_+^2, s^2, \sigma^2]^\top$, $\mathbf{n} = [\sigma_+^2, s_+^2, \sigma^2, s^2]^\top$. This formulation involves only terms about derivatives of given schedule functions, which brings computational convenience. Tab. 1 provides all ingredients needed to calculate curves of $\dot{\Delta}$ and $\dot{\Sigma}$ in schedules of VP, VE and EDM.

We also generalize the previous takeaways from the DDPM case study to $s$-$\sigma$ scheduled setting with the following analyses and discussion.

*$\sigma$ is better for the design of sampling and weighting than $s$.* It stands due to its direct reflection about SNR (signal-to-noise ratio), and additionally, because $s$ is usually adapted to heuristic motivations.

Table 1: The ingredients of generalized curves $\dot{\Delta}$ and $\dot{\Sigma}$ schedules about mainstream SDE designs, including VP, VE (Song et al., 2021), EDM (Karras et al., 2022).

| Schedules | $s$ | $\sigma^2$ | $\dot{s}$ | $\dot{\sigma}$ |
|---|---|---|---|---|
| VP | $\exp\{-\frac{1}{4}\Delta_\beta t^2 - \frac{1}{2}\beta_0 t\}$ | $\exp\{\frac{1}{2}\Delta_\beta t^2 + \beta_0 t\} - 1$ | $-\frac{\sigma\dot{\sigma}}{(1+\sigma^2)^{3/2}}$ | $\frac{(1+\sigma^2)(\Delta_\beta t + \beta_0)}{2\sigma}$ |
| VE | 1 | $t$ | 0 | 1 |
| EDM | 1 | $t^2$ | 0 | $2t$ |

In DDPM, corresponding to VP in Tab. 1, the SDE design is simply from data to normal noise. In VE, realistic diffusion processes inspire that the diffusion rate is limited to $\sigma = \sqrt{t}$. Further in EDM, motivation become more complex of training objective and concise of schedule and motivation at the same time, bringing benefits to training. Its key ideas and designs are 1) the std. of inputs and targeted outputs of a neural network $F_\theta$ in EDM is constrained to 1 with preconditioning; 2) the weights $w_t$ in Eqn. 1 are allocated according to $c_{\text{out}} w_t = 1$, where $c_{\text{out}}$ is the scale factor of F-prediction neural network $F_\theta$'s outputs, and is related to $\sigma$ and the std. of data (sometimes normalized as 1).

*Sampling deserves more attention.* The SDE design goal of most diffusion models nowadays is to add much larger noise to the data so that the samples can cover larger space. However, in terms of the process increment, it always results in a low signal-to-noise ratio of the late data when $t$ is large. Either the standard deviation is too large or the $s$ used to suppress it is too small. For instance, EDM does not bias the data distribution from expectations due to the scale $\dot{\Delta} = 0$, but $\dot{\Sigma} = 2[(t+\mathbf{d}t)^2 + t^2]$ is a quadratic increase as $t$ grows. In VP, $s \to 0$, as $t$ grows, leads that the model needs to recover expectations in approximation. For these samples, which are not very informative, a single weight adjustment is not as efficient as directly reducing the sampling rate.

## 2.4 OVERVIEW OF SPEED

Based on the above observations and analyses, we propose SpeeD , a novel approach for achieving lossless training acceleration tailored for diffusion models. As illustrated in Fig. 2, SpeeD suppresses the trivial time steps from convergence area, and weight the rapid-change intervals between acceleration and deceleration areas. Correspondingly, two main modules, asymmetric sampling and change-aware weighting, are proposed. Asymmetric sampling uses a two-step step function to respectively suppress and increase the sampling probability corresponding to trivial and beneficial time steps. Change-aware weighting is based on the change rate of process increment $\partial_t \Psi(t)$.

## 2.5 ASYMMETRIC SAMPLING

SpeeD adopts the time steps sampling probability $\mathbf{P}(t)$ as the step function in Eqn. 3 to construct the loss in Eqn. 1. We first define $\tau$ as the step threshold in $\mathbf{P}(t)$. The pre-defined boundary $\tau$ means the area where the time step are suppressed. The sampling probability is $k$ times from time-steps where $t < \tau$ than that from $t > \tau$ instead of the uniform sampling $\mathbf{U}(t) = 1/T$.

$$\mathbf{P}(t) = \begin{cases} \frac{k}{T+\tau(k-1)}, & 0 < t \leq \tau \\ \frac{1}{T+\tau(k-1)}, & \tau < t \leq T \end{cases}, \quad \text{where suppression intensity } k \geq 1 \text{ and } \tau \in (0, T]. \quad (3)$$

**Threshold Selection $\tau$.** According to Theorem 1, given a magnitude $r$, $\tau$ should satisfy $\hat{r}(\tau) > r$ to make sure that $\tau > t_{\text{d-c}}$, where the time steps suppressed are all time steps in the convergence area. To maximum the number of suppressed time steps, we set $\tau \leftarrow \sqrt{2T \log r / \Delta_\beta + T^2 \beta_0^2 / \Delta_\beta^2} - T\beta_0 / \Delta_\beta$.

## 2.6 CHANGE-AWARE WEIGHTING

According to Theorem 1, a faster change of process increment means fewer samples at the corresponding noise level. This leads to under-sampling in acceleration and deceleration areas. Change-aware weighting is adopted to mitigate the under-sampling issue. The weights $\{w_t\}_{t \in [T]}$ are assigned based on the gradient of the variance over time, where we use the approximation $\partial_t \hat{\Psi}_t$ in Theorem 1.

(a) Visualization comparison on MetFaces dataset.  (b) Visualization comparison on FFHQ dataset.

Figure 4: Our SpeeD obtains significant improvements than baseline in visualizations. More visualizations on other datasets and tasks can be found in the Appendix D.

The original gradient $\partial_t \hat{\Psi}_t$ is practically not suitable for weighting due to its small scale. Therefore, $\partial_t \hat{\Psi}_t$ is re-scaled into range $[1\text{-}\lambda, \lambda]$ that $\min\{1, \max_t \partial_t \hat{\Psi}_t\} \to \lambda$ and $\max\{0, \min_t \partial_t \hat{\Psi}_t\} \to 1\text{-}\lambda$, where symmetry ceiling $\lambda \in [0.5, 1]$. $\lambda$ regulates the curvature of the weighting function. A higher $\lambda$ results in a more obvious distinction in weights between different time-steps.

## 3 EXPERIMENTS

We first present visualizations in Sec. 3.1 and describe the main experiment setups in Sec. 3.2, including benchmark datasets, network architectures, training details, and evaluation protocols. In Sec. 3.3, we present the main results regarding both performance and efficiency. After that, in Sec. 3.6 and 3.4, we ablate the effectiveness of each designed module and evaluate the generalization of SpeeD in various diffusion tasks and settings. Finally, we investigate the compatibility of our approach with other recent methods in 3.5.

### 3.1 VISUALIZATION

The comparison of visualizations between SpeeD and DiT-XL/2 models on the MetFaces and FFHQ datasets clearly demonstrates the superiority of SpeeD. As shown in Fig. 4, SpeeD achieves significantly better visual quality at just 20K or 30K training iterations, compared to DiT-XL/2. This highlights that SpeeD reaches high-quality results much faster than the baseline method, making it a more efficient and effective approach for training diffusion models.

### 3.2 IMPLEMENTATION DETAILS

**Datasets.** We mainly investigate the effectiveness of our approach on the following datasets: MetFaces (Karras et al., 2020) and FFHQ (Karras et al., 2019) are used to train unconditional image generation, CIFAR-10 (Krizhevsky et al., 2009) and ImageNet-1K (Deng et al., 2009) are used to train conditional image generation, and MS-COCO (Lin et al., 2014) is used to evaluate the generalization of our method in the text to image task. More details of these datasets can be found in the Appendix A.

**Network architectures.** U-Net (Ronneberger et al., 2015) and DiT (Peebles & Xie, 2023) are two famous architectures in the diffusion model area. We implement our approach on these two architectures and their variants. We follow the same hyper parameters as the baseline by default. More information about the details of the architectures can be found in Appendix A.3.

**Training details.** We train all models using AdamW (Kingma & Ba, 2014; Loshchilov & Hutter, 2017) with a constant learning rate 1e-4. We set the maximum step in training to 1000 and use the linear variance. All images are augmented with horizontal flip transformations if not stated otherwise. Following common practice in the generative modeling literature, the exponential moving average (EMA) (Gardner Jr, 1985) of network weights is used with a decay of 0.9999. The results are reported using the EMA model. Details can be found in Tab. 7.

**Evaluation protocols.** In inference, we default to generating 10K images. Fréchet Inception Distance (FID) is used to evaluate both the fidelity and coverage of generated images.

Table 2: The FID↓ comparison to the baseline: DiT-XL/2, re-weighting methods: P2 and Min-SNR, and re-sampling methods: Log-Normal and CLTS. All methods are trained with DiT-XL/2 for 50K iterations. We report the FID per 10K iterations. Our approach achieves the best results on MetFaces and FFHQ datasets. **Bold entries** are best results. Following previous work (Go et al., 2023), more results of 100K iterations and *longer training phase* with different schedules are in Appendix B.1.

| dataset | MetFaces | | | | | FFHQ | | | | |
|---|---|---|---|---|---|---|---|---|---|---|
| iterations | 10K | 20K | 30K | 40K | 50K | 10K | 20K | 30K | 40K | 50K |
| DiT-XL/2 (Peebles & Xie, 2023) | 398.7 | 132.7 | 74.7 | 36.7 | 29.3 | 356.1 | 335.3 | 165.2 | 35.8 | 12.9 |
| P2 (Choi et al., 2022) | 377.1 | 328.2 | 111.0 | 27.3 | 23.3 | 368.7 | 357.9 | 158.6 | 35.5 | 15.0 |
| Min-SNR (Hang et al., 2023) | 389.4 | 313.9 | 52.1 | 31.3 | 28.6 | 376.3 | 334.1 | 151.2 | 34.0 | 12.6 |
| Log-Normal (Esser et al., 2024) | **311.8** | 165.1 | 63.9 | 51.1 | 47.3 | **307.6** | 165.1 | **63.9** | 51.1 | 47.3 |
| CLTS (Xu et al., 2024) | 375.0 | 57.2 | 28.6 | 24.6 | 23.5 | 336.1 | 329.1 | 173.4 | 33.7 | 12.7 |
| SpeeD (ours) | 367.3 | **23.4** | **22.6** | **22.1** | **21.1** | 322.1 | **320.0** | 91.8 | **19.8** | **9.9** |

## 3.3 COMPARISONS WITH OTHER STRATEGIES.

**Performance Comparisons.** Before our comparison, we first introduce our baseline, *i.e.*, DiT-XL/2, a strong image generation backbone as introduced in DiT (Peebles & Xie, 2023). We follow the hyperparameter settings from DiT and train DiT-XL/2 on MetFaces (Karras et al., 2020) and FFHQ (Karras et al., 2019), respectively. We compare our approach with two re-weighting methods: P2 (Choi et al., 2022) and Min-SNR (Hang et al., 2023), and two re-sampling methods: Log-Normal (Karras et al., 2022) and CLTS (Xu et al., 2024). In the evaluation, we use 10K generated images to calculate FID (Heusel et al., 2017) for comparison. To make a detailed comparison, all the approaches are trained for 50K iterations and we report the FID scores per 10K iterations.

As shown in Tab 2, compared to DiT-XL/2, re-weighting, and re-sampling methods, our approach obtains the best FID results. Specifically, at the 50K iteration, compared to other methods, we reduce 2.3 and 2.6 FID scores on MetFaces and FFHQ at least. Another interesting finding is that the re-weighting methods reduce the FID very slowly at the beginning of the training, *i.e.*, from the 10K to 20K iterations. That aligns with our analysis well: re-weighting methods involve a lot of steps from the convergence area. Based on the experimental results, the time steps that come from the convergence area indeed contribute limited to the diffusion training.

**Efficiency comparisons.** In addition to the performance comparison, we also present the acceleration results of our SpeeD . This naturally raises a question: *how to calculate the acceleration?* Here, we follow the previous diffusion acceleration methods (Gao et al., 2023) and other efficient training papers (Qin et al., 2023; Xu et al., 2024): visualizing the FID-Iteration curve and reporting the estimated highest acceleration ratio. We mainly compare with DiT-XL/2, one re-weighting method Min-SNR (Choi et al., 2022), and one re-sampling method CLTS (Xu et al., 2024) in Fig 5. At the same training iterations, our approach achieves significantly better FID scores than other methods. Notably, SpeeD accelerates the Min-SNR, and CLTS by 2.7 and 2.6 times, respectively. More comparisons with other methods can be found in Appendix B.

For the comparison with the baseline, *i.e.*, DiT-XL/2, considering the 50K iterations might be too short for converge, we extend the training iterations from 50K to 200K. In the long-term training, we speed up the DiT-XL/2 by 4 times without performance drops. That shows the strong efficiency of our proposed method. Most importantly, we can save **3∼5** times the overall training cost with very minimal overhead. For instance, we save 48 hours (result obtained by training on 8 A6000 Nvidia GPUs) of training time for DiT-XL/2 with negligible seconds overhead.

## 3.4 GENERALIZATION EVALUATION

**Cross-architecture robustness evaluation.** There are mainly two architectures in the diffusion models: U-Net (Ronneberger et al., 2015) and DiT (Peebles & Xie, 2023). SpeeD is not correlated to specific model architecture, thereby it is a model-agnostic approach. We implement our method with DiT-XL/2 and U-Net on MetFaces, FFHQ, and ImageNet-1K, respectively. **We default to training the models for 50K iterations on MetFaces and FFHQ, 400K on ImageNet-1K.**To ensure a fair

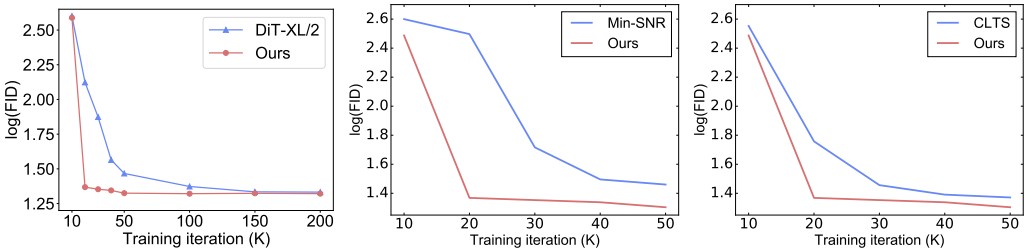

Figure 5: We plot the FID-Iteration curves of our approach and other methods on the MetFaces dataset. *SpeeD* accelerates other methods obviously. The horizontal and vertical axes represent the training iterations and log(FID), respectively.

Table 3: Cross-architecture robustness evaluation. 'Baseline' denotes training diffusion models without acceleration strategy. 'DiT' refers to the DiT-XL/2 network. All FID scores are obtained by testing 10K generated images.

| | Metfaces | | FFHQ | | ImageNet | |
| | DiT | U-Net | DiT | U-Net | DiT | U-Net |
|---|---|---|---|---|---|---|
| baseline | 36.61 | 46.77 | 12.86 | 17.37 | 26.74 | 45.71 |
| SpeeD | **21.13** | **22.88** | **9.95** | **16.52** | **20.63** | **37.33** |
| improve | 15.48 | 23.89 | 2.91 | 0.85 | 6.11 | 7.38 |

Table 4: Comparisons of FID and IS scores on FFHQ with different schedules on time steps. We mainly evaluate the generalization of our approach on linear, quadratic, and cosine schedules. We use the vanilla DiT-XL/2 as the baseline.

| | linear | | quadratic | | cosine | |
| | FID ↓ | IS ↑ | FID ↓ | IS ↑ | FID ↓ | IS ↑ |
|---|---|---|---|---|---|---|
| baseline | 12.86 | 4.21 | 11.12 | 4.21 | 18.31 | 4.10 |
| SpeeD | **9.95** | **4.23** | **9.78** | **4.29** | **17.79** | **4.15** |
| improve | 2.91 | 0.02 | 1.34 | 0.08 | 0.52 | 0.05 |

comparison, we keep all hyper-parameters the same and report the FID scores at 50K iterations. As shown in Tab. 3, SpeeD consistently achieve significantly higher performance under all settings, which indicates the strong generality of our method for different architectures and datasets.

**Cross-schedule robustness evaluation.** In the diffusion process, there are various time step schedules, including linear (Ho et al., 2020), quadratic and cosine (Nichol & Dhariwal, 2021) schedules. We verify SpeeD's robustness across these schedules. We report FID and inception score (IS) (Salimans et al., 2016) scores as metrics for comparisons. As shown in Tab. 4, SpeeD achieves significant improvement on linear, quadratic, and cosine schedules both in FID and IS. That shows the strong generality of SpeeD in various schedules.

**Cross-task robustness evaluation.** We apply SpeeD to the text-to-image task for evaluating the generality of our method. For text-to-image generation, we first introduce CLIP (Radford et al., 2021) to extract the text embedding for MS-COCO (Lin et al., 2014) dataset. Then, DiT-XL/2 is used to train a text-to-image model as our baseline. Following prior work(Saharia et al., 2022), FID score

Table 5: Text to image.

| Method | FID↓ | CLIP score↑ |
|---|---|---|
| baseline | 27.41 | 0.237 |
| SpeeD | **25.30** | **0.244** |

and CLIP score on MS-COCO validation set are evaluation metrics for quantitative analyses. As illustrated in Tab. 5, we obtain the better FID and CLIP score than our baseline.

### 3.5 COMPATIBILITY WITH OTHER ACCELERATION METHODS

Until now, we evaluate the effectiveness and generalization of our proposed method: SpeeD is a task-agnostic and architecture-agnostic diffusion acceleration approach. Is SpeeD compatible with other acceleration techniques? To investigate this, we apply our approach with two recent proposed algorithms: masked diffusion transformer (MDT) (Gao et al., 2023) and fast diffusion model (FDM) (Wu et al., 2023).

**MDT + SpeeD.** MDT (Gao et al., 2023) proposes a masked diffusion transformer method, which applies a masking scheme in latent space to enhance the contextual learning ability of diffusion probabilistic models explicitly. MDT can speed up the diffusion training by 10 times. They evaluate their MDT with DiT-S/2. We just inject our SpeeD on their MDT and report FID-Iteration curves for comparison in Fig. 6a. All the results are obtained on ImageNet-1K dataset. Our approach can *further* accelerate MDT at least by **4** ×, which indicates the good compatibility of SpeeD.

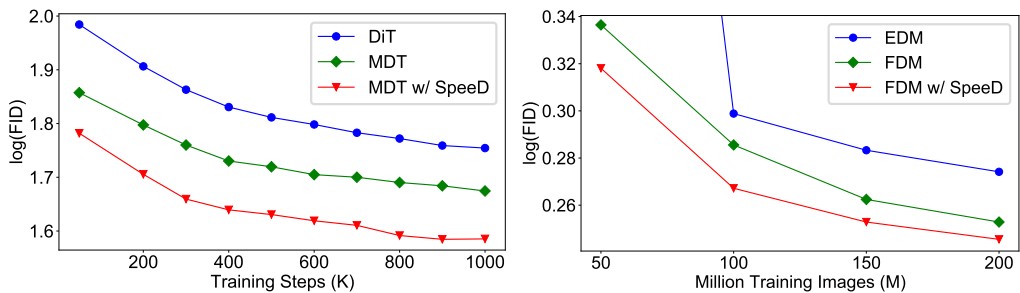

(a) FID-Iteration curve comparisons on *ImageNet-1K*.    (b) FID-Iteration curve comparisons on *CIFAR-10*.

Figure 6: SpeeD works well with recent acceleration algorithms and can consistently further accelerate the diffusion model training. We plot the figures using $\log(\text{FID})$ for better visualization.

Table 6: Ablation studies on FFHQ dataset. Default settings and baseline are in gray and cyan.

(a) **Components of SpeeD**. Suppressing some trivial time steps does help.

(b) **Suppression intensity** $k$. Huge suppression decrease diversity to modeling data.

(c) **Symmetry ceiling** $\lambda$. Weighting served as temperature factor should be in moderation.

| sampling curve | CAW | FID ↓ |
|---|---|---|
| uniform | | 17.37 |
| uniform | ✓ | 16.75 |
| asymmetric | | 15.82 |
| asymmetric | ✓ | **15.07** |

| suppression intensity $k$ | FID ↓ |
|---|---|
| 1 | 15.01 |
| 5 | **14.86** |
| 10 | 16.97 |
| 25 | 25.59 |

| symmetry ceiling $\lambda$ | FID ↓ |
|---|---|
| 0.5 | 15.46 |
| 0.6 | **14.86** |
| 0.8 | 16.83 |
| 1.0 | 23.77 |

**FDM + SpeeD.** Fast Diffusion Model (Wu et al., 2023) is a diffusion process acceleration method inspired by the classic momentum approach to solve optimization problem in parameter space. By integrating momentum into the diffusion process, it achieves similar performance as EDM (Karras et al., 2022) with less training cost. Based on the official implementation, we compare EDM, FDM, and FDM + SpeeD on CIFAR-10 of $32 \times 32$ images. FDM accelerates the EDM by about $1.6 \times$. Meanwhile, SpeeD can further reduce the overall training cost around $1.6 \times$.

### 3.6 ABLATION EXPERIMENTS

We perform extensive ablation studies to illustrate the characteristics of SpeeD . The experiments in ablation studies are conducted on the FFHQ dataset and U-Net model by default. We ablate our designed components: asymmetric sampling (abbreviated as asymmetric) and change-aware weighting (abbreviated as CAW), suppression intensity $k$ in asymmetric sampling defined in Eqn. 3 and symmetry ceiling $\lambda$ for weighting in Sec. 2.6.

**Evaluating the components of SpeeD.** Our approach includes two strategies: asymmetric sampling and change-aware weighting. We note these two strategies using 'asymmetric' and 'CAW'. We ablate each component in SpeeD. As illustrated in Tab. 6a, combining our proposed strategies achieves the best results. Using weighting and sampling strategies alone improves the baseline by 0.6 and 1.5 FID scores, respectively, indicating that filtering most samples from the convergence area is more beneficial for training.

**Evaluating of suppression intensity $k$.** To prioritize the training of critical time steps, asymmetric sampling focus on the time steps out of the convergence area. A higher probability is given for these time steps which is $k$ times than the time steps from the convergence area. A larger suppression intensity $k$ means a larger gap in training between the different areas of time steps. We evaluate different suppression intensity $k$ from 1 to 25 and report the FID score In Tab. 6b. We observe that k = 5 achieves the best performance. A huge suppression intensity decrease FID scores seriously, which means that it hurts the diversity a lot to modeling data. This means that the samples in the convergence area, although very close to pure noise, still contains some useful information. Extreme complete discard of these samples results in a degradation of the acceleration performance.

**Evaluating of symmetry ceiling $\lambda$.** The symmetry ceiling $\lambda$ is a hyper-parameter that regulates the curvature of the weighting function. $\lambda$ **is set in the interval** $[0.5, 1]$. The midpoint of the re-scaled weight interval is fixed at $0.5$. The symmetry ceiling $\lambda$ is the right boundary of the interval and the left boundary is 1-$\lambda$. A higher $\lambda$ results in a larger weight interval and a more obvious distinction in weights between different time steps. In Tab. 6c, settings $\lambda \leq 0.8$ obtain higher performance on FID scores than the baseline, which indicates SpeeD is relatively robust on symmetry ceiling $\lambda$. Further increase $\lambda$ leads to performance degradation. This means weighting should be in moderation.

## 4 RELATED WORK

We discuss the related works about Diffusion Models and its Training Acceleration. The most related works are as follows, and more about cross-modality and video generation are in Appendix C.

**Diffusion models**  Diffusion models have emerged as the dominant approach in generative tasks (Saharia et al., 2022; Chen et al., 2023b; Wang et al., 2024a; Igashov et al., 2024), which outperform other generative methods including GANs (Zhu et al., 2017; Isola et al., 2017; Brock et al., 2018), VAEs (Kingma & Welling, 2013), flow-based models (Dinh et al., 2014). These methods (Ho et al., 2020; Song et al., 2020; Karras et al., 2022) are based on non-equilibrium thermodynamics (Jarzynski, 1997; Sohl-Dickstein et al., 2015), where the generative process is modeled as a reverse diffusion process that gradually constructs the sample from a noise distribution (Sohl-Dickstein et al., 2015). Previous works focused on enhancing diffusion models' generation quality and alignment with users in visual generation. To generate high-resolution images, Latent Diffusion Models (LDMs) (Saharia et al., 2022; Rombach et al., 2022) perform diffusion process in latent space instead of pixel space, which employ VAEs to be encoder and decoder for latent representations.

**Acceleration in diffusion models**  To reduce the computational costs, previous works accelerate the diffusion models in training and inference. For *training acceleration*, the earliest works (Choi et al., 2022; Hang et al., 2023) assign different weights to each time step on Mean Square Error (MSE) loss to improve the learning efficiency. The other methods in training acceleration are proposed, *e.g.*, network architecture (Ryu & Ye, 2022; Wang et al., 2024b) and diffusion algorithm (Karras et al., 2022; Wu et al., 2023). Masking modeling (Gao et al., 2023; Zheng et al., 2023) are recently proposed to train diffusion models. Works (Go et al., 2023; Park et al., 2024b;a; Kim et al., 2024) provide observations for explanation from the perspective of multi-tasks learning. SpeeD is of good compatibility with these methods, *e.g.*, (Yu et al., 2024; Zheng et al., 2024a;b). In the field of sampling acceleration, a number of works tackle the slow inference speed of diffusion models by using fewer reverse steps while maintaining sample quality, including DDIM (Song et al., 2020), Analytic-DPM (Bao et al., 2022), and DPM-Solver (Lu et al., 2022).

**Conditional generation.**  To better control the generation process with various conditions, *e.g.*, image style, text prompt and stroke, Classifier-Free Guidance (CFG) proposes a guidance strategy with diffusion models that balance the sample quality and prompt alignment. ControlNet (Zhang et al., 2023b) reuses the large-scale pre-trained layers of source models to build a deep and strong encoder to learn specific conditions. Recently benefiting from diffusion models in the image generation field, video generation (Ma et al., 2024; Lab & etc., 2024) is getting trendy.

## 5 CONCLUSION

We present SpeeD , a novel approach for accelerating diffusion training by closely examining time steps. The core insights from this examination are: 1) suppressing the sampling probabilities of time steps that offer limited benefits to diffusion training (i.e., those with extremely small losses), and 2) emphasizing the importance of time steps with rapidly changing process increments. SpeeD demonstrates strong robustness across various architectures and datasets, achieving significant acceleration on multiple diffusion-based image generation tasks. Additionally, SpeeD is easily compatible with other diffusion acceleration methods, highlighting its wide applicability. We provide extensive theoretical analysis in this paper, aiming to support future research in both academia and industry.

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

## A  More Detail of Experiments

In this section, we introduce detailed experiment settings, datasets, and architectures.

### A.1  Statement

Experimental design note: Referring to previous works (Peebles & Xie, 2023; Go et al., 2023; Park et al., 2024b; Choi et al., 2022; Hang et al., 2023; Esser et al., 2024; Xu et al., 2024), our experiments train 50K, 100K, and 400K in MetFaces $256 \times 256$, FFHQ $256 \times 256$, and ImageNet-1K, respectively. we know that DiT used 7M iterations to train ImageNet in the original work Peebles & Xie (2023), which 1) is not the amount of resource usage that can be achieved in general research. Meanwhile, 2) the focus of this paper is on the acceleration effect of training, and a direct comparison of the ultimate convergence stage is very unnecessary for the research topic of this paper.

### A.2  Instructions for code

We have submitted the source code as the supplementary materials in a zipped file named as 'SpeeD.zip' for reproduction.

### A.3  Architecture and Training Recipe.

We utilize Unet and DiT as our base architecture in the diffusion model. pre-trained VAE which loads checkpoints from huggingface is employed to be latent encoder. Following Unet implementation from LDM and DIT from official implementation, we provide the architecture detail in Tab. 7. We provide our basic training recipe and evaluation setting with specific details in Tab. 8.

| Architecture | input size | input channels | patch size | model depth | hidden size | attention heads |
|---|---|---|---|---|---|---|
| U-Net | $32 \times 32$ | 3 | - | 8 | 128 | 1 |
| DiT-XL/2 | $32 \times 32$ | 4 | 2 | 28 | 1152 | 16 |
| DiT-S/2 | $32 \times 32$ | 4 | 2 | 12 | 384 | 6 |

Table 7: Architecture detail of Unet and DiT on MetFaces, FFHQ, and ImageNet.

| | MetFaces $256 \times 256$ | FFHQ $256 \times 256$ | ImageNet-1K $256 \times 256$ |
|---|---|---|---|
| latent size | $32 \times 32 \times 3$ | $32 \times 32 \times 3$ | $32 \times 32 \times 3$ |
| class-conditional | | | ✓ |
| diffusion steps | 1000 | 1000 | 1000 |
| noise schedule | linear | linear | linear |
| batch size | 256 | 256 | 256 |
| training iterations | 50K | 100K | 400K |
| optimizer | AdamW | AadamW | AdamW |
| learning rate | 1e-4 | 1e-4 | 1e-4 |
| weight decay | 0 | 0 | 0 |
| sample algorithm | DDPM | DDPM | DDPM |
| number steps in sample | 250 | 250 | 250 |
| number sample in evaluation | $10,000$ | $10,000$ | $10,000$ |

Table 8: Our basic training recipe based on MetFaces, FFHQ, ImageNet datasets

### A.4  Datasets

**CIFAR-10**. CIFAR-10 datasets consist of $32 \times 32$ size colored natural images divided into categories. It uses $50,000$ in images for training and EDM evaluation suite in image generation.

**MetFaces** is an image dataset of human faces extracted from works of art. It consists of 1336 high-quality PNG images at 1024×1024 resolution. We download it at 256 resolution from kaggle.

**FFHQ** is a high-quality image dataset of human faces, contains 70,000 images. We download it at 256x256 resolution from kaggle.

**ImageNet-1K** is the subset of the ImageNet-21K dataset with $1,000$ categories. It contains $1,281,167$ training images and $50,000$ validation images.

**MSCOCO** is a large-scale text-image pair dataset. It contains 118K training text-image pairs and 5K validation images. We download it from official website.

**FaceForensics** is a video dataset consisting of more than 500,000 frames containing faces from 1004 videos that can be used to study image or video forgeries.

### A.5 DETAIL OF MDT + SPEED EXPERIMENT

MDT utilizes an asymmetric diffusion transformer architecture, which is composed of three main components: an encoder, a side interpolater, and a decoder. During training, a subset of the latent embedding patches is randomly masked using Gaussian noise with a masking ratio. Then, the remaining latent embedding, along with the full latent embedding is input into the diffusion model.

Following the official implementation of MDT, We utilize DiT-S/2 and MDT-S/2 as our base architecture, whose total block number both is 12 and the number of decoder layers in MDT is 2. We employ the AdamW (Loshchilov & Hutter, 2017) optimizer with constant learning rate 1e-4 using 256 batch size without weight decay on class-conditional ImageNet with an image resolution of $256^2$. We perform training on the class-conditional ImageNet dataset with images of resolution 256x256. The diffusion models are trained for a total of 1000K iterations, utilizing a mask ratio of 0.3.

### A.6 DETAIL OF FDM + SPEED EXPERIMENT

FDM add the momentum to the forward diffusion process with a scale that control the weight of momentum for faster convergence to the target distribution. Following official implementation, we train diffusion models of EDM and FDM. We retrain these official network architecture which is U-Net with positional time embedding with dropout rate 0.13 in training. We adopt Adam optimizer with learning rate 1e-3 and batch size 512 to train each model by a total of 200 million images of $32^2$ CIFAR-10 dataset. During training, we adopt a learning rate ramp-up duration of 10 Mimgs and set the EMA half-life as 0.5 Mimgs. For evaluation, EMA models generate 50K images using EDM sampler based on Heun's $2^{nd}$ order method (Süli & Mayers, 2003).

### A.7 TEXT-TO-IMAGE EXPERIMENT DETAIL

In text to image task, diffusion models synthesize images with textual prompts. For understanding textual prompts, text-to-image models need semantic text encoders to encode language text tokens into text embedding. We incorporate a pre-trained CLIP language encoder, which processes text with a maximum token length of 77. DiT-XL/2 is employed as our base diffusion architecture. We employ AdamW optimizer with a constant learning rate 1e-4 without weight decay. We train text-to-image diffusion models for 400K training iterations on MS-COCO training dataset and evaluate the FID and CLIP score on MS-COCO validation dataset. To enhance the quality of conditional image synthesis, we implement classifier-free guidance with 1.5 scale factor.

### A.8 THEORETICAL ANALYSIS

#### A.8.1 NOTATIONS

In this section, we will introduce the main auxiliary notations and the quantities that need to be used. The range of schedule hyper-parameter group $\{\beta_t\}_{t\in[T]}$ turns out to be $t=1$ to $t=T$. For analytical convenience, we define $\beta_0$ as $\beta_0 := \beta_1 - \Delta_\beta/T$.

Another auxiliary notation is forward ratio $\rho_t$, which is defined as $\rho_t = t/T$. Forward ratio provide an total number free notation for general diffusion process descriptions.

Based on the two auxiliary notations $\beta_0$ and $\rho_t$, the expression of $\beta_t$ with respect to the forward process ratio is $\beta_t = \beta_0 + \Delta_\beta\rho_t$.

The relationship between $\alpha_t$ and $\beta_t$ is recalled and re-written as follows: $\alpha_t = 1 - \beta_t = 1 - \beta_0 - \Delta_\beta\rho_t$. $\bar{\alpha}_t$ the multiplication of $\alpha_t$ is re-written as $\bar{\alpha}_t = \Pi_{s=1}^t(1 - \beta_0 - \Delta_\beta\rho_s)$.

Perturbed samples' distribution: $x_t|x_0 \sim \mathcal{N}(\sqrt{\bar{\alpha}_t}x_0, (1-\bar{\alpha}_t)\mathbf{I})$

### A.8.2 AUXILIARY LEMMA AND CORE THEOREM

**Lemma 1** (Bounded $\alpha$ by $\beta$). *In DDPM ([Ho et al., 2020](#)), using a simple equivariant series $\{\beta_t\}_{t\in[T]}$ to simplify the complex cumulative products $\{\bar{\alpha}_t\}_{t\in[T]}$, we obtain the following auxiliary upper bound of $al\bar{p}ha_t$.*

$$\bar{\alpha}_t \leq \exp\{-(\beta_0 t + \frac{\Delta_\beta t^2}{2T})\}$$

### A.8.3 PROPOSITIONS

**Proposition A.1** (Jensen's inequality). *If $f$ is convex, we have:*
$$\mathbf{E}_X f(X) \geq f(\mathbf{E}_X X).$$
*A variant of the general one shown above:*
$$||\sum_{i\in[N]} x_i||^2 \leq N \sum_{i\in[N]} ||x_i||^2.$$

**Proposition A.2** (triangle inequality). *The triangle inequality is shown as follows, where $||\cdot||$ is a norm and $A, B$ is the quantity in the corresponding norm space:*
$$||A+B|| \leq ||A|| + ||B||$$
.

**Proposition A.3** (matrix norm compatibility). *The matrix norm compatibility, $A \in \mathbb{R}^{a\times b}, B \in \mathbb{R}^{b\times c}, v \in \mathbb{R}^b$:*
$$||AB||_m \leq ||A||_m ||B||_m$$
$$||Av||_m \leq ||A||_m ||v||.$$
**Proposition A.4** (Peter Paul inequality).

$$2\langle x, y \rangle \leq \frac{1}{\epsilon}||x||^2 + \epsilon ||y||^2$$

.

### A.8.4 PROOF OF LEMMA 1

*Proof.* To proof the auxiliary Lemma 1, we re-arrange the notation of $\bar{\alpha}_t$ as shown in Section A.8.1, and we have the following upper bound:

$$\log \bar{\alpha}_t = \sum_{s=1}^{t} \log(1 - \beta_0 - \Delta_\beta \rho_s)$$

$$\leq t \log(\frac{1}{t}\sum_{s=1}^{t}(1 - \beta_0 - \Delta_\beta \rho_s))$$

$$= t \log(1 - \beta_0 - \Delta_\beta \frac{1}{t}\sum_{s=1}^{t}\frac{s}{T})$$

$$= t \log(1 - \beta_0 - \Delta_\beta \frac{t+1}{2T})$$

$$\leq -(\beta_0 t + \frac{\Delta_\beta(t+1)t}{2T}),$$

where the two inequalities are by the concavity of log function and the inequality: $\log(1+x) \leq x$. Taking exponents on both sides simultaneously, we have:

$$\bar{\alpha}_t \leq \exp\{-(\beta_0 t + \frac{\Delta_\beta t^2}{2T})\}.$$

$\square$

### A.8.5 PROOF OF THEOREM 1

Before the proof of the theorem, we note that the samples $x_t|x_0 \sim \mathcal{N}(\mu_t, \sigma_t)$ have the following bounds with Lemma 1:

- Reformulate the expression of $\sqrt{\bar{\alpha}}$, we have the mean vector $\mu_t$'s components $\dot{\mu}_t$ bounded by $\dot{x}_0$ the corresponding components of data $x_0$ as follows:

$$\dot{\mu}_t = \sqrt{\bar{\alpha}_t}\dot{x}_0 \leq \exp\{-\frac{1}{2}(\beta_0 t + \frac{\Delta_\beta t^2}{2T})\}\dot{x}_0,$$

- Reformulate the expression of $\bar{\alpha}$, we have a partial order relation on the cone about covariance matrix of $x_t|x_0$ as follows:

$$\sigma_t = (1 - \bar{\alpha}_t)\mathbf{I} \succeq (1 - \exp\{-(\beta_0 t + \frac{\Delta_\beta t^2}{2T})\})\mathbf{I}.$$

*Proof.* The process increment at given $t^{\text{th}}$ time step is $\delta_t = x_{t+1} - x_t$. $\delta_t$ is a Gaussian process as follows:

$$\delta_t \sim \mathcal{N}(\underbrace{(\sqrt{\alpha_{t+1}} - 1)\sqrt{\bar{\alpha}_t}x_0}_{\phi_t}, \underbrace{[2 - \bar{\alpha}_t(1 + \alpha_{t+1})]\mathbf{I}}_{\Psi_t})$$

The theorem's key motivation is that the label is noisy, and noisy magnitude is measured by mean vector's norm $||\phi_t||$ and covariance matrix $\Psi_t$.

The upper bounds of mean vectors' norm and the partial order of covariance matrix at different time step $t$ are shown as follows:

$$||\phi_t||^2 \leq (\sqrt{\alpha_{t+1}} - 1)^2 \bar{\alpha}_t ||\mathbb{E}x_0||^2$$
$$\leq (1 - \alpha_{t+1})\bar{\alpha}_t ||\mathbb{E}x_0||^2$$
$$\leq (\underbrace{\beta_0 + \Delta_\beta \rho_{t+1}}_{\beta_{t+1}}) \exp\{-(\underbrace{\beta_0 + \frac{\Delta_\beta t}{2T}}_{\beta_{t/2}})t\}||\mathbb{E}x_0||^2$$
$$\leq \beta_{\max} \exp\{-(\underbrace{\beta_0 + \frac{\Delta_\beta t}{2T}}_{\beta_{t/2}})t\}||\mathbb{E}x_0||^2$$

where the inequalities are by Lemma 1, $(1-x)^2 \leq (1-x^2) = (1-x)(1+x)$, when $x \in [0, 1]$, and $\beta_{t+1} \leq \beta_{\max}$

$$\Psi_t = [2(1 - \bar{\alpha}_t) + \bar{\alpha}_t(\beta_0 + \Delta_\beta \rho_{t+1})]\mathbf{I}$$
$$\succeq 2(1 - \exp\{-(\underbrace{\beta_0 + \frac{\Delta_\beta t}{2T}}_{\beta_{t/2}})t\})\mathbf{I} + \bar{\alpha}_t\beta_{t+1}\mathbf{I}$$
$$\succeq 2(1 - \exp\{-(\underbrace{\beta_0 + \frac{\Delta_\beta t}{2T}}_{\beta_{t/2}})t\})\mathbf{I}$$

where the inequalities are by Lemma 1 and $\bar{\alpha}_t\beta_{t+1}\mathbf{I} \succeq \mathbf{0}$. The residual term is

$$\bar{\alpha}_t\beta_{t+1} = \beta_{t+1}\Pi_{s=1}^t(1 - \beta_s) \geq \exp\{\log \beta_{t+1} + t\log(1 - \beta_t)\}$$

$\square$

## B MORE EXPERIMENT RESULTS

**Efficiency comparisons.** In Fig. 7, besides the Min-SNR and CLTS, we show the efficiency comparison with P2 and Log-Normal methods. One can find that our method consistently accelerates the diffusion training in large margins.

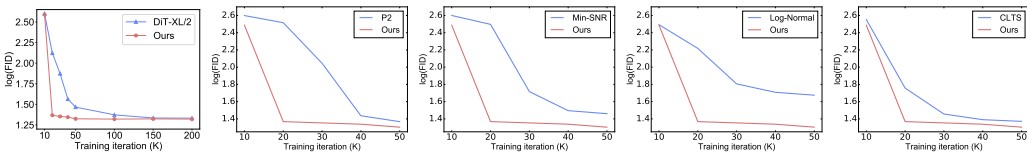

Figure 7: More efficiency comparison on MetFaces.

**Super resolution with SpeeD.** We employ SpeeD to super-resolution image generation on $512 \times 512$ MetFaces compared with vanilla DiT. We train DiT-XL/2 for 100K training iterations and compare the FID score at 50K, 100K training iterations. The batch size is 32 for saving the GPU memory. As shown in 9, SpeeD obtain better performance than vanilla DiT at same training iterations on $512^2$ MetFaces dataset. It indicates that SpeeD can achieve training acceleration on super-resolution tasks.

Table 9: Super resolution.

| Method | 50K | 100K |
|--------|------|------|
| DiT-XL/2 | 77.9 | 35.4 |
| SpeeD | 48.7 | 10.6 |

### B.1 ADDITIONAL EXPERIMENTS

#### B.1.1 100K ITERATIONS CUT ON FFHQ

A cut of training process comparison on FFHQ dataset are shown in Tab. 10.

| Schedules | linear | quadratic | cosine |
|-----------|--------|-----------|--------|
| DiT-XL/2 (Peebles & Xie, 2023) | 7.8 | 7.6 | 8.0 |
| ANT-UW (Go et al., 2023) | 6.3 | 7.1 | 8.3 |
| TskDif-Cur (Park et al., 2024b) | 6.7 | 6.7 | 7.2 |
| P2 (Choi et al., 2022) | 10.1 | 10.2 | 12.3 |
| MinSNR (Hang et al., 2023) | 7.6 | 7.6 | 11.2 |
| Log-Normal (Esser et al., 2024) | 15.6 | 14.9 | 19.2 |
| CLTS (Xu et al., 2024) | 7.9 | 7.7 | 9.1 |
| SpeeD(ours) | **5.8** | **5.8** | **6.4** |

Table 10: 100K iterations cut on FFHQ.

#### B.1.2 GUIDANCE SCALE

Ablation experiments about guidance scale in related conditional generation task are shown in Tab. 11.

| Guidance | 0.75 | 1 | 1.5 | 2 | 3 |
|----------|------|------|------|------|------|
| t2i DiT-XL/2 | 20.1 | 22.5 | 27.4 | 31.0 | 36.2 |
| t2i SpeeD | 19.4 | 20.8 | 25.3 | 28.8 | 34.8 |
| ccg EDM | 1.8 | 1.8 | 1.9 | 1.9 | 2.5 |
| ccg SpeeD on EDM | 1.7 | 1.7 | 1.8 | 1.8 | 2.3 |

Table 11: Text-to-image (t2i) task on MS-COCO at 400K iteration and EDM on CIFAR-10 with 200M training images for class-conditional generation (ccg).

#### B.1.3 DIT ARCHITECTURE

Ablation experiments about DiTs of different architectures are shown in Tab. 12.

|      | S/2  | B/2  | XL/2 |
|------|------|------|------|
| DiTs | 18.1 | 12.9 | 7.8  |
| SpeeD| 15.3 | 10.8 | 5.8  |

Table 12: Different size of DiT with 100K iterations on FFHQ.

| Iterations (K) | 10    | 20    | 30    | 40   | 50   | 60   | 70   | 80  | 90  | 100 |
|----------------|-------|-------|-------|------|------|------|------|-----|-----|-----|
| DiT-XL/2       | 356.1 | 335.3 | 165.2 | 35.8 | 12.9 | 11.9 | 10.5 | 9.6 | 8.7 | 7.8 |
| SpeeD          | 322.1 | 320.0 | 91.8  | 19.8 | 9.9  | 7.6  | 7.1  | 6.6 | 6.2 | 5.8 |

Table 13: Details about training to 100K on FFHQ.

### B.1.4 DETAILED TRAINING PROCESS

The detailed training process on FFHQ through 100K iterations are shown in Tab. 13.

## C MORE RELATED WORKS

We discuss other works related to SpeeD, including Text to Image and Video generation. Another point to mention is that we learn from InfoBatch (Qin et al., 2023) in writing.

**Text to image generation with diffusion models** Text-to-image generation has emerged as a hotly contested and rapidly evolving field in recent years, with an explosion of related industrial products springing up (Saharia et al., 2022; Rombach et al., 2022; Betker et al., 2023; Chen et al., 2023a; Esser et al., 2024). Convert textual descriptions into corresponding visual content, models not only learn to synthesize image content but also ensuring alignment with the accompanying textual descriptions. To better align images with textual prompt guidance, previous work has primarily focused on enhancements in several schemes including strengthening the capacity of text encoder (Raffel et al., 2020; Radford et al., 2021) improving the condition plugin module in diffusion model (Zhang et al., 2023b), improving data quality (Betker et al., 2023).

**Video generation with diffusion models.** As diffusion models achieve tremendous success in image generation, video generation has also experienced significant breakthroughs, marking the field's evolution and growth. Inspired by image diffusion, pioneering works such as RVD Yang et al. (2022) and VDM Ho et al. (2022b) explore video generation using diffusion methods. Utilizing temporal attention and latent modeling mechanisms, video diffusion has advanced in terms of generation quality, controllability, and efficiency Ho et al. (2022a); Singer et al. (2022); Zhou et al. (2022); He et al. (2022); Zhang et al. (2023a); Guo et al. (2023); Wu et al. (2022); Wang et al. (2023). Notably, Stable Video Diffusion Blattmann et al. (2023) and Sora Brooks et al. (2024) achieve some of the most appealing results in the field.

**Other diffusion acceleration works** To achieve better results with fewer NFE steps, Consistency Models Song et al. (2023) and Consistency Trajectory Models Kim et al. (2023) employ consistency loss and novel training methods. Rectified Flow Liu et al. (2022), followed by Instaflow Liu et al. (2023), introduces a new perspective to obtain straight ODE paths with enhanced noise schedule and improved prediction targets, together with the reflow operation.

## D VISUALIZATION

**Comparisons to the baseline and other methods.** The Fig. 8 compares our method (Ours) with the baseline and other acceleration methods (P2 and Min-SNR) in terms of FID scores at various training iterations (K). The baseline method starts with a high FID score of 335.2 and gradually decreases to 12.8, showing slow convergence and less sharp final images. P2 begins with a slightly higher FID score of 357.9 and reduces to 15.0, improving faster than the baseline but still exhibiting slower convergence compared to Min-SNR and our method. Min-SNR starts with 334.1 and achieves a final score of 12.2, producing clearer and higher quality images consistently compared to the baseline

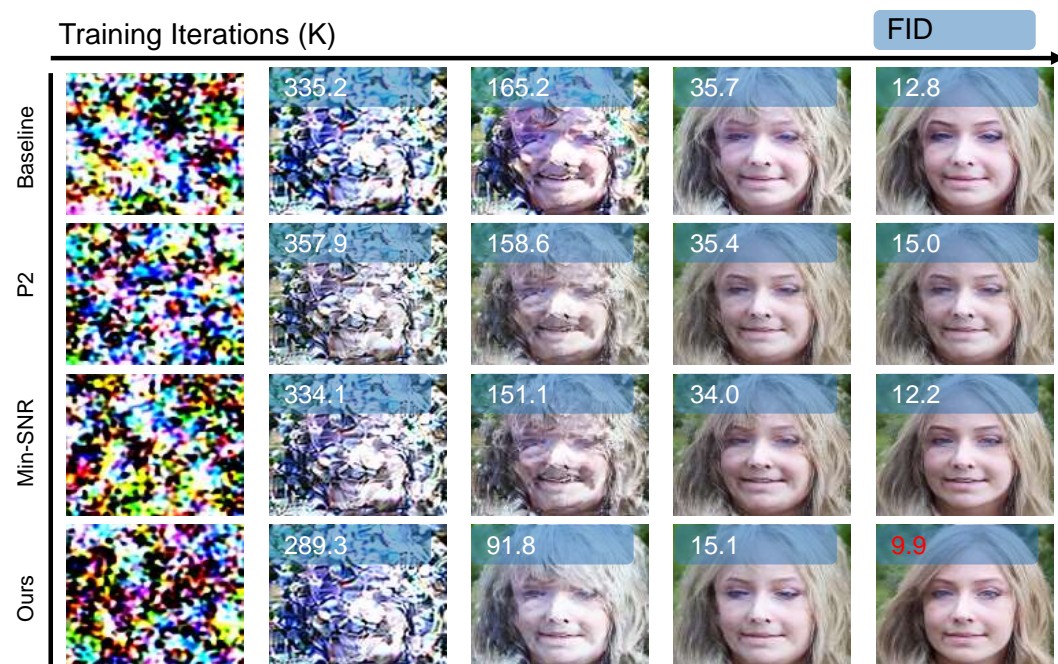

Figure 8: Comaprisons to the baseline and other acceleration methods.

and P2. Our method stands out, starting with a significantly lower FID score of 289.3 and rapidly dropping to 9.9, demonstrating the fastest convergence and the best overall image quality. This rapid decrease highlights the efficiency of our approach, requiring fewer iterations to produce high-quality results. Overall, our method outperforms the baseline, P2, and Min-SNR in both speed and quality of image generation, proving its effectiveness in accelerating the training of diffusion models.

**Visualizations of the generated images.** The figures above illustrate the quality of images generated by our method across various datasets, including CIFAR-10, FFHQ, MetFaces, and ImageNet-1K. In Fig. 9, the generated images from the CIFAR-10 dataset display distinct and recognizable objects, even for challenging categories. Fig. 10 presents generated images from the FFHQ dataset, showcasing diverse and realistic human faces with varying expressions and features. Fig. 11 exhibits images from the MetFaces dataset, depicting detailed and lifelike representations of artistic portraits. Finally, Fig. 12 includes images from the ImageNet-1K dataset, featuring a wide range of objects and scenes with excellent accuracy and visual fidelity. These results emphasize the superior performance of our method in generating high-quality images across different datasets, indicating its potential for broader applications in image synthesis and computer vision tasks.

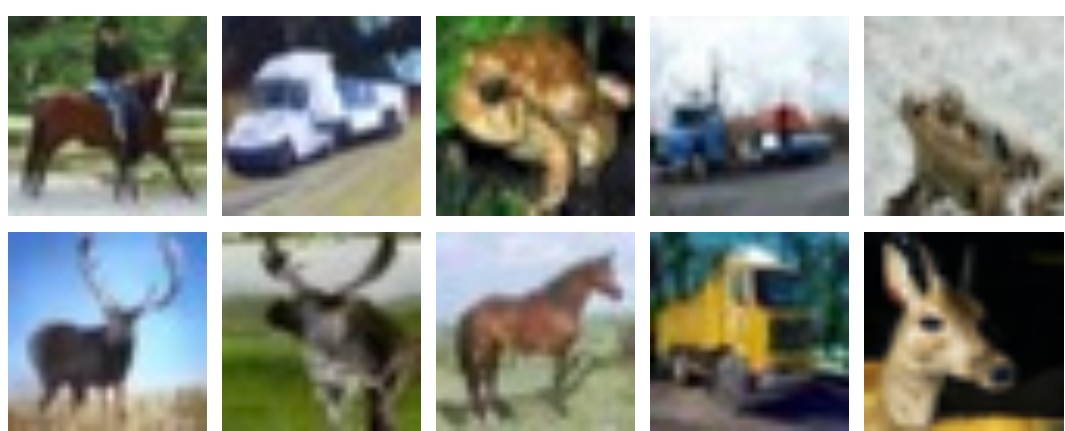

Figure 9: Generated images of CIFAR-10.

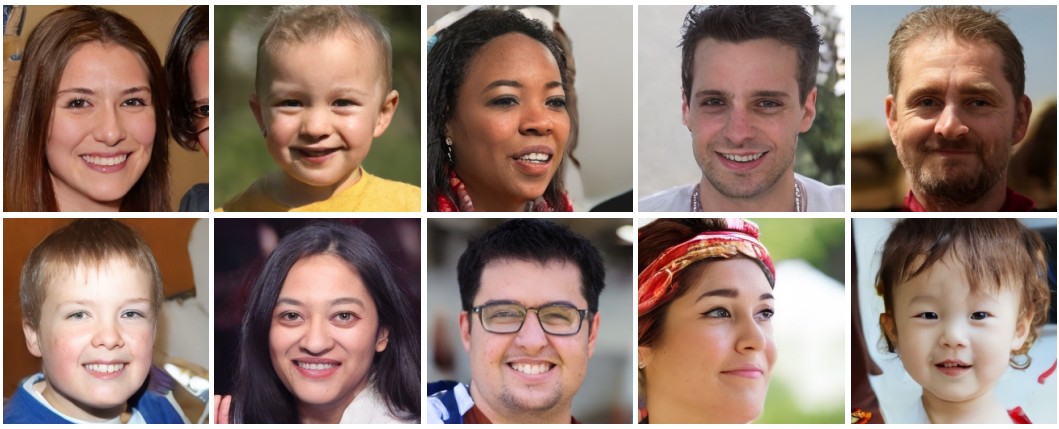

Figure 10: Generated images of FFHQ.

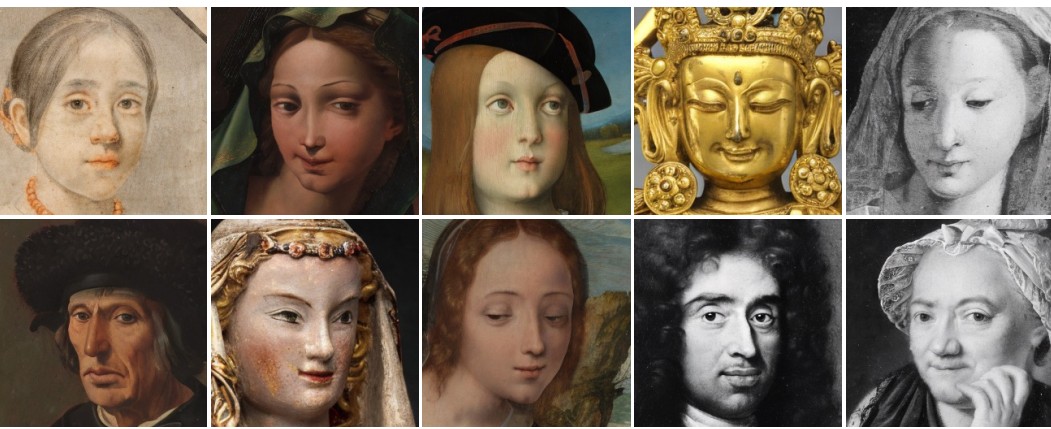

Figure 11: Generated images of MetFaces.

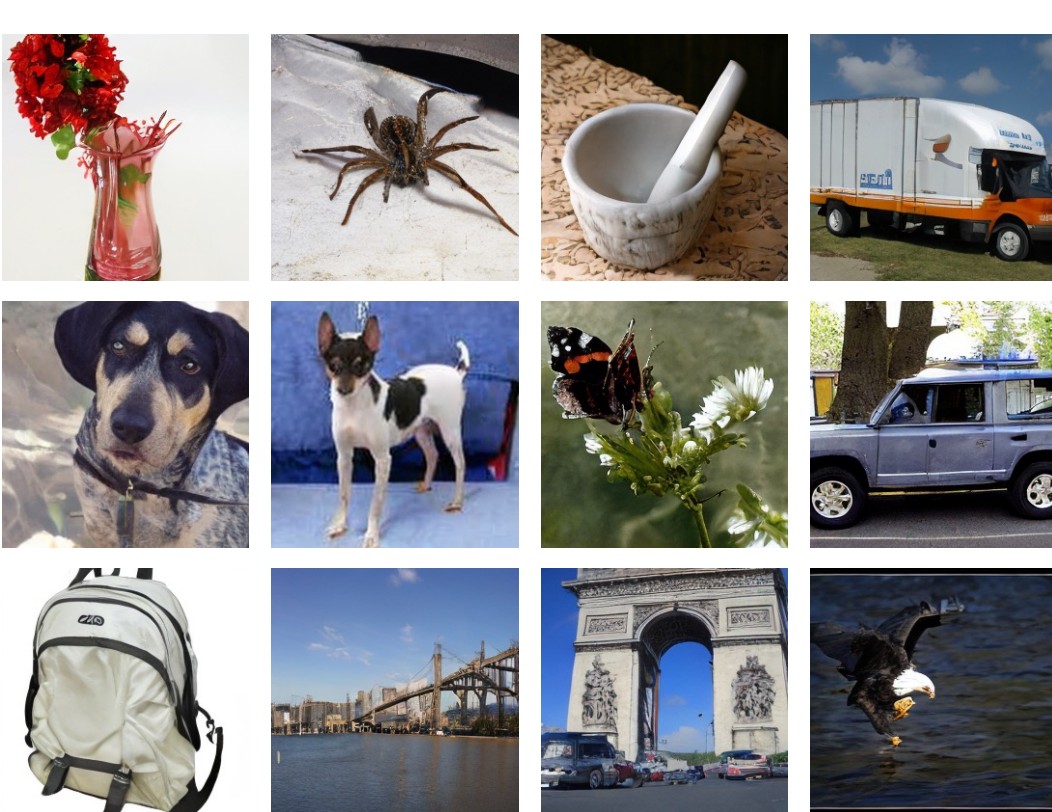

Figure 12: Generated images of ImageNet-1K.

