# OpenReview forum: "A Closer Look at Time Steps is Worthy of Triple Speed-Up for Diffusion Model Training"
_ICLR.cc/2025/Conference — ICLR 2025 Conference Withdrawn Submission_

### Official Review · Reviewer_nX3R · 2024-10-29

**Soundness:** 3
**Presentation:** 2
**Contribution:** 2
**Rating:** 3
**Confidence:** 4

**Summary:**

This paper proposes to speed up the training process of the diffusion model with an adaptive sampling strategy. The training process is
 empirically divided into 3 stages based on increment: acceleration, deceleration, and convergence. Then the sampling strategy will reduce the frequency of steps from the convergence area. The importance of time steps is also considered. Five baselines are introduced compared with the proposed method on 2 datasets. Overall, this paper studies an important issue of diffusion models. However, there are major flaws and the experiment quality does not allow the acceptance of this paper.

**Strengths:**

1. The long training process of the diffusion model is a critical issue for computational cost.
2. A scheduling mechanism is proposed to dynamically to adjust the sampling strategy.
3. The proposed method is evaluated on two datasets by comparing it with several baselines.

**Weaknesses:**

1. The rationale behind the motivation is not clearly stated and verified. The author states that the time steps can be empirically divided into 3 states. However, there is no empirical result to support the claim. All figures in the method part (i.e., Figure 1 and Figure 2) are pseudo figures. The value in real experiments should be provided.

2. It is not practical to decide the boundary of each state in real application. Also, it is not very clear how to decide the boundary of each state. If the depends on the convergent speed, the states could vary significantly based on the learning rate, model framework, and the quality of training data. Such a strategy is not practical. In fact, the analysis is not comprehensive. Is it possible that an adaptive learning rate will address the issue? The author should exclude other factors to verify it is the sampling quality, not another factor that results in the difference between the 3 stages.

3. The whole paper assumes that the diffusion model is DDPM. However, there are so many papers[1] that have already addressed the quality of sampling such as DDIM. The author should address the problem with a SOTA framework regarding efficiency.

4. The quality of the experiment is low. While the motivation is to speed up the training process, is it intuitive to report the real training time? The majority of the experiment is report FID with baselines. FID is not the metric to verify the efficiency. Figures 5 and 6 are used to report the convergent speed. However, it looks Log(FID) is not convergent yet. Also, what is the learning rate for each baseline in Figure 5? Why there are 3 sub-figures in Figure 5? Should it be a single figure including a comparison with all baselines?

5. Ablation study is missing. The author should remove the sampling strategy for each stage and vary the boundary. The presentation in the experiment could be improved.

6. Important baselines are missing including [2,3,4, 5] and many others.

[1] Shivam Gupta, Ajil Jalal, Aditya Parulekar, Eric Price, Zhiyang Xun:
Diffusion Posterior Sampling is Computationally Intractable.
[2] Tae Hong Moon, Moonseok Choi, EungGu Yun, Jongmin Yoon, Gayoung Lee, Jaewoong Cho, Juho Lee:
A Simple Early Exiting Framework for Accelerated Sampling in Diffusion Models. ICML 2024
[3] Zhiwei Tang, Jiasheng Tang, Hao Luo, Fan Wang, Tsung-Hui Chang:
Accelerating Parallel Sampling of Diffusion Models. ICML 2024
[4] Towards Faster Training of Diffusion Models: An Inspiration of A Consistency
Phenomenon
[5] Hongkai Zheng, Weili Nie, Arash Vahdat, Kamyar Azizzadenesheli, Anima Anandkumar:
Fast Sampling of Diffusion Models via Operator Learning. ICML 2023: 42390-42402
[6] 	Andy Shih, Suneel Belkhale, Stefano Ermon, Dorsa Sadigh, Nima Anari:
Parallel Sampling of Diffusion Models. NeurIPS 2023

Overall, there are major drawbacks in the proposed method and the quality of the experiment can be significantly improved.

**Questions:**

Please refer to the weakness.

---

### Official Review · Reviewer_z27S · 2024-11-02

**Soundness:** 3
**Presentation:** 2
**Contribution:** 3
**Rating:** 5
**Confidence:** 3

**Summary:**

The paper studies an important problem of accelerating diffusion model training. The authors proposed a division of timesteps into three categories of acceleration/deceleration/convergence. The author provides evidence that the convergence region is of no big importance to the training, and thus propose an asymmetric sampling strategy SpeeD to focus on the earlier timesteps. Multiple empirical results are shown to demonstrate the effectiveness of the proposed methodology.

**Strengths:**

This paper made several interesting contributions. Although there are other works which studies the importance of timesteps in diffusion model training, the analysis in the division of timesteps in this paper seems novel and technically solid. This novel observation then leads to a simple asymmetric sampling and reweighting approach. The paper is overall well-organized and well-presented. Many experiments are conducted to demonstrate the effectiveness of the proposed method, and the results are quite convincing. Finally, as the proposed methodology is architecture-agnostic and can be used in a plug-and-play manner in addition to many other acceleration methods, it could potentially be very versatile and be widely used.

**Weaknesses:**

1. Although the paper is overall well-presented and well-motivated, the presentation of some sections can be made clearer. For example, could the authors elaborate on the rescaling scheme in section 2.6?
2. The authors provide ablation studies showcasing the transferability of SpeeD into other types of schedule. However, it seems that the improvement regarding training speed for the quadratic and cosine schedulers are not as strong compared to linear scheduler. As the theoretical analysis is solely based on linear scheduler, can similar results be provided in the quadratic or cosine scheduler case?
3. Contrary to previous works, the authors argues one should put higher probability in sampling earlier timesteps (up to a threshold $\tau$) rather than the middle timesteps. Could the authors provide some empirical ablations on the choice of the threshold $\tau$?

**Questions:**

See the weakness section, and also the following.

1. In the current approach, the asymmetric sampling does not differentiate between the accelerate and decelerate region and put both of them to high higher probability compared to the convergence region. However, I believe these two regions are differentiated in the reweighting part. Could the authors clarify if this is the case and provide a more thorough discussion on the reweighting strategy? Please also refer to weakness (1).
2. From table 3 in paper, it seems that the proposed method's effectiveness is not as significant when evaluated on FFHQ compared to Metfaces, could the author provide intuition why?

---

### Official Review · Reviewer_h2ZU · 2024-11-03

**Soundness:** 2
**Presentation:** 2
**Contribution:** 2
**Rating:** 3
**Confidence:** 4

**Summary:**

The article proposes a methodology to accelerate training in diffusion models. Their proposal is presented in the context of other acceleration methods and builds on an analysis of the way training is performed in DMs. The proposed method is validates empirically in a number of experiments.

**Strengths:**

The experimental validation of the article seems thorough. The proposed method is compared against several acceleration methods, and an ablation study is performed.

**Weaknesses:**

I have to admit that I found this paper hard to follow.

The paper is redundant and lacks proper organisation. For instance, on the last page of the article, just before the conclusions, the authors revisit what a diffusion model is (while the entire paper is about DMs) and mention classifier guidance (which is not referred to in the paper whatsoever).

The diagrams in the first pages (Figs 1,2 & 3) do not really help to clarify the contribution. They are cluttered, and it is difficult to understand what they are trying to illustrate. It is unclear whether these are based on data or just illustrations. In any case, it is unclear what the authors mean when the paper refers to these figures stating that they help to "visualize a loss curve"

The detailed presentation of the proposed method is also unclear. The first 4 pages of the paper introduce the motivation and context, and back up the observations leading to the proposed method. However, in Sec 2.4 (?), the method is finally presented but without the necessary clarity. Is eq (3) the main definition of the procedure?

Sec 2 is 3.5 pages long and covers the basics of DM and the brief presentation (see above) of the method. Then, the paper jumps directly to the experiments.

Overall, I think that there is certainly some practical value in this article, but there a fair amount of work needed in term of presentation and organisation to take this paper at the acceptance bar at ICLR.

**Questions:**

please see above

---

### Official Review · Reviewer_uk6P · 2024-11-04

**Soundness:** 3
**Presentation:** 4
**Contribution:** 2
**Rating:** 5
**Confidence:** 5

**Summary:**

This paper introduces SpeeD, a novel approach aimed at improving training efficiency for diffusion models. By analyzing the process increment between adjacent timesteps, the authors categorize timesteps into acceleration, deceleration, and convergence zones based on process increment bounds. Through asymmetric sampling and change-aware weighting, SpeeD effectively reduces the number of timesteps required for training, as demonstrated by comprehensive experimental results.

**Strengths:**

1. The paper addresses a critical issue—accelerating diffusion model training—and includes comparisons with existing methods.
2. The proposed analysis of process increment bounds is both novel and theoretically grounded.
3. The experiments are thorough and persuasive, showcasing the effectiveness of the proposed method.

**Weaknesses:**

1. **Unclear motivation:** The paper introduces the analysis of $ \delta_t = x_{t+1} - x_t $ as a representation of each timestep, yet the motivation behind this choice could be clearer. Furthermore, the authors use $ \partial_t \hat{\Psi}_t $ for timestep categorization. While $ \delta_t $, representing the amount of change, seems more intuitive, the motivation for using $ \partial_t \hat{\Psi}_t $ lacks explanation. Additional insights into this choice would enhance the understanding of the approach.

2. **Limited applicability to general schedules:** The method relies on a linear schedule for discrete-timestep diffusion models. Although general cases are mentioned, extending SpeeD to other scenarios, such as EDM, and supporting this with experiments would strengthen the paper’s applicability.

**Questions:**

1. **How do the re-sampling and weighting strategies differ?**
   Re-sampling and weighting strategies share a similar motivation and produce identical effects in the objective function. Existing studies typically focus on either re-sampling or weighting, while SpeeD combines both approaches. Could the authors elaborate on the rationale for using both strategies instead of just one, and provide experimental evaluations to highlight the differences?

---

### Note · Authors · 2024-11-14

**Comment:**

There is still room for improvement, thanks to reviewers for the valuable comments.

**Withdrawal Confirmation:**

I have read and agree with the venue's withdrawal policy on behalf of myself and my co-authors.